# Acute Abdominal Pain: Missed Diagnoses, Extra-Abdominal Conditions, and Outcomes

**DOI:** 10.3390/jcm9040899

**Published:** 2020-03-25

**Authors:** Isabelle Osterwalder, Merve Özkan, Alexandra Malinovska, Christian H. Nickel, Roland Bingisser

**Affiliations:** Department of Emergency Medicine, University Hospital Basel, 4051 Basel, Switzerland; isabelle.osterwalder@usb.ch (I.O.); merve.oezkan@hotmail.com (M.Ö.); alexandra.malinovska@usb.ch (A.M.); christian.nickel@usb.ch (C.H.N.)

**Keywords:** abdominal pain, emergency department, missed diagnoses, extra-abdominal causes of abdominal pain

## Abstract

Abdominal pain (AP) is a common reason for presentation to an emergency department (ED). With this prospective, observational all-comer study, we aimed to answer three questions: Which diagnoses are most often missed? What is the incidence of extra-abdominal causes? What is the prognosis of abdominal pain in a tertiary urban European ED? Participants were systematically interviewed for the presence of 35 predefined symptoms. For all patients with abdominal pain, the index visit diagnoses were recorded. Related representation was defined as any representation, investigation, or surgery related to the index visit (open time frame). If a diagnosis changed between index visit and representation, it was classified as missed diagnosis. Among 3960 screened presentations, 480 (12.1%) were due to AP. Among 63 (13.1%) related representations, the most prevalent causes were cholelithiasis, gastroenteritis, and urinary retention. A missed diagnosis was attributed to 27 (5.6%) presentations. Extra-abdominal causes were identified in 162 (43%) presentations. Thirty-day mortality was comparable to that of all other ED patients (2.2% vs. 2.1%). Patients with abdominal pain had a low risk of representation, and the majority of representations due to missed diagnoses were of benign origin. The high incidence of extra-abdominal causes is noteworthy, as this may induce change to differential diagnosis of abdominal pain.

## 1. Introduction

Abdominal pain (AP) is among the most common reasons to present to an emergency department (ED) [1,2,3]. In the US and in Europe, it is consistently one of the top chief complaints [4,5]. Differential diagnosis ranges between self-limiting nonspecific abdominal pain [6,7] and life-threatening conditions [8,9,10]. Abdominal pain is a notoriously difficult symptom [11,12] due to diagnostic uncertainty [13] and the risk of representation [14].

It has been described that abdominal pain is amongst the most frequent problems associated with malpractice claims at the emergency department [15]. Other issues are the occurrence of unexpected diagnoses [16] of extra-abdominal origin [17] and problematic “off-hour” presentations with abdominal pain. The influence of age regarding the diagnosis and prognosis of abdominal pain has not been well described in the literature.

Therefore, three questions can be considered unanswered or controversial. First: Which diagnoses are most often missed and lead to representation? Second: What is the prevalence of extra-abdominal causes of abdominal pain? Third, what is the short- and long-term prognosis of abdominal pain in a tertiary urban European ED, stratified according to age?

We performed a prospective, observational study with a one-year follow-up, designed as an all-comer analysis in which all patients were interviewed at presentation to the ED by a study team. To our knowledge, there are no data on the true prevalence of abdominal pain in the ED, as the cited studies have all relied on the determination of a chief complaint by triage nurses or emergency physicians. As there is evidence that “off-hour” patients differ in acuity [18] and that symptoms may be filtered and selected when attributing “chief complaints” [19], we interviewed patients in a highly standardized fashion at presentation and around the clock, in order to minimize selection bias.

We hypothesized that missed diagnoses leading to representation are considerable (>10%), that extra-abdominal causes of abdominal pain are frequent (>25%), and that prognosis is highly dependent on age in this notoriously difficult patient population.

## 2. Methods

### 2.1. Study Design

This prospective monocentric all-comer study was conducted in the emergency department of the University Hospital of Basel, a tertiary care center in Northwestern Switzerland with an annual ED census of more than 50,000 patients and a capacity of about 700 beds. The investigation took place within two time periods lasting three weeks each. The first one started on 21 October and ended on 11 November 2013, the second one was realized between 1 and 23 February 2015. Two different time points were chosen because of logistic and financial reasons and to minimize seasonal effects by choosing two different seasons. The local ethics committee (EKNZ, Basel, Switzerland, www.eknz.ch) approved the conduct of the study (Project no. 236/13, 8 October 2013).

### 2.2. Selection of Participants

Every patient presenting to the ED during the study periods was eligible. Pediatric and obstetric patients presenting to facilities nearby were not included. Patients undergoing life-saving interventions and patients who were unconscious, intoxicated, or could not be interviewed due to mental issues were not included. Multiple presentation was not excluded. Patients gave informed consent in order to participate. The Emergency Severity Index (ESI) was used to triage all patients, level 1 being the highest acuity and urgency, and level 5 the lowest [20].

### 2.3. Data Collection

A dedicated study team, comprised of medical students, was trained to collect data using a standardized questionnaire. They worked in three shifts to include patients 24 h a day, 7 days per week. When presenting to the ED, patients were registered, and an electronic health record (EHR) was generated. They were triaged by a nurse or a physician according to the German version of the ESI [20]. Participants were systematically interviewed for the presence of the following 35 symptoms: skin rash, headache, dizziness, acute visual problem, acute hearing problem, feeling feverish, rhinorrhea, dysphagia, cough, expectoration, dyspnea, chest pain, abdominal pain, nausea, vomiting, diarrhea, constipation, dysuria, back pain, neck pain, arm pain, leg pain, joint pain, flank pain, joint swelling, leg swelling, altered mentation, numbness, paralysis, gait disorders, speech disorders, fatigue, weakness, lack of appetite, feeling sleepy. The data were checked by an external institution (Health Care Research Institute, Zürich, Switzerland), digitalized, and anonymized.

All patients’ charts were abstracted by two independent chart reviewers. Medical record review was performed with the EHR database (ISMed; ProtecData AG, Boswil, Switzerland). A case report form, created with Microsoft Access 2016, was used, in order to document the results. To classify diagnoses, a three-digit ICD-10 (International Statistical Classification of Diseases and Related Health Problems, version from 2019) code was generated by the two independent chart reviewers. The patients’ underlying conditions were taken from the discharge report. In case of disagreement between the two chart reviewers, an experienced emergency physician served as a referee. The main diagnoses at the index visit were recorded in the database after agreement of all reviewers.

In the event of a representation, the diagnoses at representation were coded independently of the index visit in a second step, in order to minimize anchoring bias. In a third step, the diagnoses were compared between index visit and representation. Related representation was defined as any representation, investigation, or surgery in relation to the index visit (e.g., another presentation to the emergency department or any outpatient visit due to AP). There was no time frame limiting a related representation, and chart abstraction to assess follow-up was performed over one year after the index visit. Two reviewers independently decided if the representation was related to the index visit. If there was a change of diagnosis between index visit and related representation, the case was classified as missed diagnosis. In case of discordant classifications of the reviewers, the referee was called as described above. According to our expert group a “low rate” was defined as less than 10% missed diagnoses. A “considerable rate” was defined as over 10% missed diagnoses associated with representations. These definitions are expert opinion only, and hypotheses were made by the study group after broad assessment of the literature in this field.

We defined abdominal diagnoses as diseases originating in the peritoneal, retro-peritoneal (except for renal and urinary tract system), and pelvic compartments, including vascular problems. This includes diseases of the digestive system, coded in chapter K in the ICD-10-system, and other diseases of abdominal origin, such as malignancies (chapter C), diseases of abdominal vessels (chapter I), or infectious diseases (chapter A/B) of abdominal origin. Further, diseases originating in the retro-peritoneal (pancreas) or pelvic compartments (reproductive system) were also subsumed to abdominal diagnoses. Viral, infectious, and unspecified gastroenteritis were condensed (A08, A09, K29), as well as urinary tract infection and cystitis (N30, N39). We defined extra-abdominal diagnoses as diseases causing abdominal pain not originating in the abdominal or pelvic compartment. This included diagnoses of all other compartments or organ systems, such as chest (e.g., pneumonia, myocardial infarction), urogenital (e.g., urolithiasis, urinary retention), cerebral (e.g., epileptic manifestations), blood (e.g., sickle cell crisis), as well as psychiatric diagnoses (e.g., somatoform disorders), intoxications, or systemic diseases (e.g., porphyria, diabetes) [21]. After a literature review, our experts decided to use the term “frequent” for extra-abdominal causes of AP, if >25% of the diagnoses were due to an extra-abdominal origin. Eleven out of 12 criteria of Worster’s methods [22] were used for chart reviews. Blinding to the hypotheses was impossible, as reviewers were aware of the study objectives.

Information about length of stay (LOS), as well as hospitalization outcomes, intensive care unit (ICU) transfer, and mortality was extracted from the EHR. One year after presentation, a follow-up regarding survival was conducted. Data were extracted from the EHR, from official residents’ and insurance registries, as well as directly from patients, proxies, and family physicians.

Hospitalization, according to the Swiss law, was defined as at least one overnight stay in a hospital bed. LOS was defined as the number of days spent in hospital during the index hospitalization. ICU transfer was defined as any admission to medical, surgical, or neurosurgical ICUs or to a stroke unit during the index hospitalization. In-hospital mortality was defined as the percentage of patients who died after presenting to the ED without being discharged between admission and death. Thirty-day and one-year mortality were defined as the percentage of deceased patients at 30 days and 1 year after presenting to the ED. 

If patients presented more than once during the inclusion periods, we did not exclude them and treated them as independent presentations for all analyses, except for mortality. Only the first presentation was taken for mortality calculations.

### 2.4. Outcomes

The rate of missed diagnoses, the incidence of extra-abdominal conditions, and the prognosis, as assessed by hospitalization, ICU transfer, and mortality, were determined.

### 2.5. Statistical Analyses

Descriptive statistics are presented as counts and frequencies for categorical data and medians (first quartile, third quartile) for metric variables. Overall p-values correspond to the Mann–Whitney U test for median and to the chi-squared or exact Fisher test when the expected frequencies were less than 5. A *p*-value <0.05 was considered significant. All calculations were performed using the statistical software R (version 3.5.0, Foundation for Statistical Computing, Vienna, Austria).

## 3. Results

### 3.1. Study Enrolment

During the study period, 5634 presentations were registered. In total, 4703 cases were eligible for screening, and 3960 (100%) were included for further analyses. Among these, 480 (12.1%) presented with AP, and 3480 (87.9%) presented with other symptoms. We registered 310 outpatient presentations and 170 inpatient presentations out of 480 AP presentations. Within the group of 310 presentations, 250 represented to our hospital’s ED or outpatient clinics, 60 did not represent in the open time period after the index visit. Among 250 representations, 63 were related to the index presentation, and 187 presentations were caused by other health problems; 27 “missed diagnoses” were found among the 63 related representations. AP was the seventh most prevalent symptom in this all-comer population (see Figure 1).

The median age of AP patients was 47 (32, 68) years, and the median age of patients lacking AP was 51 (33, 72) years. A statistically significant higher number of patients (53%) were female among the AP patients, as compared to 48% female patients in the non-AP group. AP patients were attributed to a statistically significant higher triage level (*p* < 0.001) (for details, see Table 1).

### 3.2. Diagnoses

Patients presenting with abdominal pain received more than 150 different final diagnoses according to three-digit ICD codes. Gastroenteritis was the most common diagnosis (*n* = 60, 10.8%), followed by nonspecific abdominal pain (NSAP, *n* = 58, 10.4%) and cholelithiasis (*n* = 25, 4.5%). In patients <65 years old, gastroenteritis (*n* = 54, 15.1%) and NSAP (*n* = 53, 14.1%) headed the list, followed by urolithiasis (*n* = 22, 5.8%). Patients ≥65 years old were most likely to suffer from diverticulitis (*n* = 13, 7.3%), cholelithiasis (*n* = 10, 5.6%), and urinary retention (*n* = 9, 5.0%) (for details, see Table 2).

### 3.3. Representations

In total, 250 (80%) of the 310 outpatients represented. Among these, 63 cases (25%) were classified as related representations. Patients with related representations presented earlier (median 5 days (2, 25)) than patients presenting with an unrelated problem (median 162 days (935, 407)). Almost 90% of all related representations took place within 50 days after the index visit (see Figure 2).

Among the 63 related representations, the most prevalent causes were cholelithiasis (*n* = 8, 13%) followed by gastroenteritis (*n* = 7, 11%) and urinary retention (*n* = 4, 6%). A missed diagnosis was attributed to 27 (5.6%) cases. Eight (1.7%) of all AP patients underwent surgery after missed diagnoses (for details, see Table 3).

### 3.4. Extra-Abdominal Diagnoses

In total, 162 (43%) extra-abdominal diagnoses were made, the most frequent ones being of urogenital and pulmonary origin. Nineteen (5%) AP presentations were due to pulmonary problems in younger patients, and 20 (11.2%) in patients 65 years old and older. Psychiatric diagnoses (F-codes according to ICD-10) were made in 17 (4.5%) cases in younger patients, as compared to 3 (1.7%) cases in older patients. Three different pulmonary entities were among the top 20 underlying conditions. Six (1.6%) AP presentations were due to underlying cancer in younger patients, as compared to nine (5%) in older patients. Of all 15 presentations due to cancer, 4 were due to newly diagnosed cancer, and 11 were due to relapse, deterioration, or complications of previously known cancer. Twenty diagnoses were responsible for 63% of all diagnoses (see Table 2).

### 3.5. Prognosis

Of all AP presentations, 170 (35.4%) patients were hospitalized, and 310 (64.6%) patients received outpatient treatment. Hospitalization in patients ≥ 65 years of age was comparable between patients with AP and without AP (83 and 683 patients, respectively (59% for both subsets)). AP patients < 65 years of age were significantly more often hospitalized compared to non-AP patients (26% versus 17%).

Median LOS among all patients presenting with AP was 5 (2.5, 9) days, also comparable to that of patients not presenting with AP. Older patients had a non-significant higher risk of ICU transfer: 10 (7.1%) versus 122 (10.5%) in the AP and non-AP subset, respectively.

In-hospital mortality was low for patients < 65 years of age; one (0.3%) patient with AP and 6 (0.3%) patients without AP died. In-hospital mortality was higher in older patients: 5 (3.6%) patients with AP and 43 (3.7%) patients without AP died.

One-year mortality among patients ≥ 65 years of age was 21 (15.3%) for patients with AP and 170 (15%) for patients without AP. Differences between the AP and non-AP subgroups regarding LOS, ICU transfer, and mortality were not statistically significant (for details, see Table 4). Finally, 6% of patients were lost to follow-up.

## 4. Discussion

The main findings of this study were the considerable rate (13.1%) of related representations, the low rate (5.6%) of missed diagnoses, the very low rate of missed diagnoses with subsequent surgery (1.7%), the high incidence (43%) of extra-abdominal causes, and the large differences in the prognosis and distribution of disease between younger and older patients.

Overall, 27 (5.6%) presentations received missed diagnoses. However, we had no evidence for missed diagnoses in hospitalized patients. Therefore, discharged patients were affected in 8.7%. Diagnoses with an over 50% likelihood of being missed at the index visit were malignancies, food intolerance, and gastroenteritis. Every second malignancy was diagnosed at the index visit in the ED, while the other half was made after referral to an outpatient clinic and was therefore counted as representation. The rate of missed diagnoses was low in our cohort, in spite of an overall high rate of representation and an excellent follow-up of over 90%, as compared to other cohorts examining unscheduled returns to the ED [23]. However, there are very few studies on missed diagnoses in patients with abdominal pain, and we are not aware of any data from the last decade. An older study reported false-negative evaluations only in ED patients with early appendicitis or small bowel obstruction [24]. The low rate of missed diagnoses could be due to the high availability of sonography and computed tomography and the standardized work-up endorsed in our hospital.

As prospective cohorts with abdominal pain in the ED tend to suffer from inclusion bias, because of a tendency to both focus on “acute abdomen” [25,26] and assess the performance of AP biomarkers [27], there are no comparable data on all-comer populations. Therefore, the question on the prevalence of extra abdominal causes of abdominal pain needs to be highlighted. As ED work-up is protocol-based, a major prerequisite for the content and the validation of such protocols is the prevalence of abdominal and extra-abdominal conditions, e.g., it is still debated if coronary heart disease (CHD) is to be ruled out in patients presenting with AP. Some ED protocols suggest routine testing for troponin, leading to serial analyses and further work-up particularly in older patients. However, troponin is only to be used as a diagnostic tool if the prevalence of CHD is considerable, our study lacking evidence for such routine testing. Two other prospective studies [28,29] have evaluated over 2000 patients with abdominal pain and, together, they have only identified one myocardial infarction and two cases of congestive heart failure. While these studies reported a prevalence of less than 5% extra-abdominal causes, we found over 40% extra-abdominal causes in patients presenting with AP. The high incidence of extra abdominal causes could influence protocol-based care. In our cohort, nearly half of the top 20 diagnoses were not of abdominal origin. Apart from pulmonary infections [30], influenza and sepsis should be actively sought for, and abdominal pain may be a sign of urolithiasis. Generally, older patients may present with atypical symptoms of common disorders [31,32]. Rather than, e.g., CHD, urinary retention or other disorders of the urinary tract are to be included in the broader differential diagnosis of abdominal pain. Particularly in younger patients, somatoform disorders have been claimed to be a leading cause of AP [33]. These entities are frequently seen in the context of psychosocial disturbances such as abuse, anxiety, depression, or personal losses [34,35]. In our cohort, somatoform disorders and functional intestinal disorders taken together were diagnosed in less the 2.5% of all younger AP patients. However, under-diagnosis is frequent in these cases, and NSAP may have been chosen instead. However, the incidence of NSAP was comparable between our cohort and other cohorts. Interestingly, there were only two representations after an initial diagnosis of NSAP.

The prognostic value of the presence of abdominal pain in ED patients was not shown in our all-comer cohort: survival, intensive care, and use of resources were comparable to those of a general ED population. While survival was excellent in patients under 65 years of age, it was lower in patients 65 and older. Abdominal pain as a group does not seem to be a high-risk symptom, unlike dyspnea, altered mentation, and nonspecific complaints, such as generalized weakness [5].

Another controversial issue is the question of “difficult diagnosis”. While the diagnosis of cholelithiasis used to be difficult before ultrasound became widely available, vascular catastrophes were often missed before CT scans were available. According to our data, urogenital problems seem to be the most likely cause of a related representation.

In comparison, the incidence of AP was comparable to that found in European [4] and US studies [36,37]. However, in our prospective all-comer cohort, AP was only the seventh most prevalent symptom, while it was the second most prevalent in the CHARITEM study and often among the top five symptoms in European studies [5]. As we systematically interviewed all patients about the presence of abdominal pain, underreporting is highly unlikely. However, over-reporting is conceivable, as more than one symptom could be chosen. In fact, the majority of our patients reported more than one symptom [38], the number of symptoms being unrelated to medically important outcomes. To our knowledge, no studies have reported age-stratified prevalence of underlying diseases in the last two decades. While the most prevalent subgroup presenting with abdominal pain are women under 65 years of age, as previously shown [39], men over 65 years of age have another distribution of underlying conditions. While nonspecific abdominal pain has the highest pre-evaluation probability in younger patients, as reported [39,40,41], we found a lower incidence, in both younger and older patients, than reported in retrospective studies [24,40,42]. This could be due to the fact that imaging is now used much more frequently than in previous years [2].

In older patients, diverticular disease was the most prevalent. A high prevalence of this disease has previously been reported [43], but in the “classic” cohorts [44,45] other conditions were more frequent, such as cholecystitis, NSAP, appendicitis, small bowel obstruction, and pancreatitis. Certain conditions seem to have become rare, such as complicated hernias and ischemic problems, in spite of the aging population.

Taken together, all categories of diagnosis may be missed and may lead to representations. However, the rate of representation leading to a modification of diagnosis and surgery was low. Further, the incidence of extra-abdominal causes in abdominal pain was higher than expected. While CHD may be less prevalent than expected, (pulmonary) infections ranging from influenza and COPD to sepsis are to be considered, according to our results. Short- and long-term prognosis of patients with abdominal pain is favorable and largely depends on the patients’ age. In the older population, differential diagnosis may even be broader; in this respect, urinary retention should be identified early on, as its prognosis is dreary [46].

This study shows that awareness of the most common extra-abdominal conditions causing AP (in our case, urogenital and pulmonary diseases) may facilitate a fast and accurate diagnosis and therapy. To improve the clinical practice, a consistent routine during the diagnostic process should be implemented in order to cover a wide range of differential diagnoses.

## 5. Conclusions

In our setting, AP is common at the ED and lead to a wide range of diagnoses. Missed diagnoses are uncommon, but extra-abdominal causes need to be considered. Therefore, protocol-based work-up needs to cover a considerable number and range of extra-abdominal conditions.

**Limitations:** Several limitations of this study need to be discussed. First, this was a single-center study, and external validity is therefore limited. However, our ED cohort seems to represent the population of other urban, European EDs, with over two-thirds of patients having their origin in central or northern Europe [5] and the proportion of foreigners in our ED being about 32% [38]. An inclusion bias is possible, because almost 17% of patients could not be screened, and 6% were lost to follow-up. In addition, patients with persistent symptoms after ED discharge could have presented to another institution.

We focused on a limited number of predefined symptoms that were actively asked for by medical students, and certain presentations may have been missed. The highly standardized assessment of 35 predefined symptoms may have influenced the results, as the typically applied “physician filter” did not come into play.

## Figures and Tables

**Figure 1 jcm-09-00899-f001:**
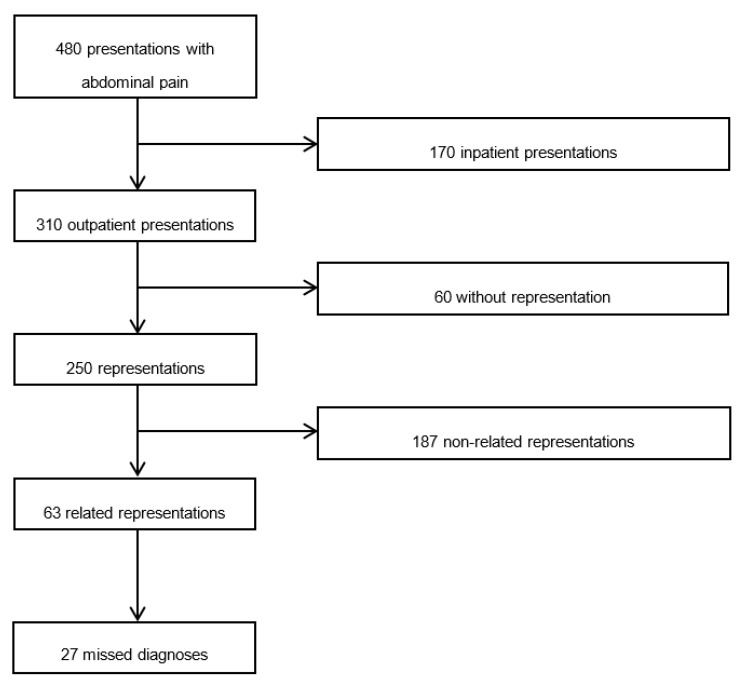
Study population.

**Figure 2 jcm-09-00899-f002:**
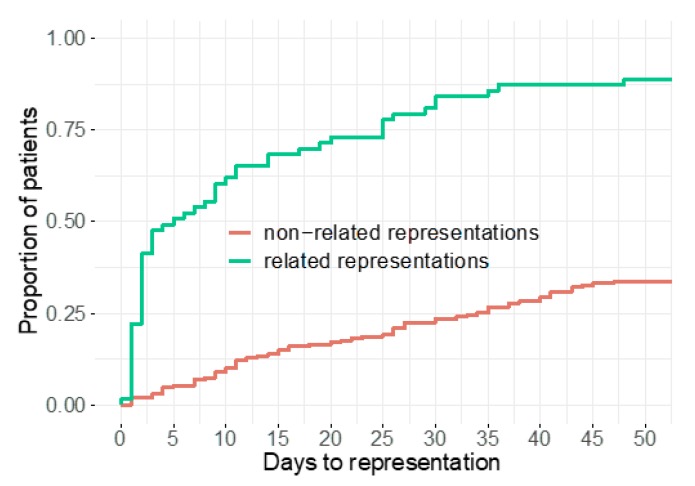
Time to representation.

**Table 1 jcm-09-00899-t001:** Baseline characteristics.

	All (*n* = 3960)		<65 Years (*n* = 2652)		≥65 Years (*n* = 1308)	
	Abdominal Pain	No Abdominal Pain	Abdominal Pain	No Abdominal Pain	Abdominal Pain	No Abdominal Pain
Presentations, n (%)	480 (12.1)	3480 (87.9)	339 (12.8)	2313 (87.2)	141 (10.8)	1167 (89.2)
Age (years), median (Q1, Q3)	47 (32, 68)	51 (33, 72)	39 (27, 49)	39 (28, 51)	77 (71, 84)	78 (72, 85)
Sex (female), n (%)	253 (52.7) *	1659 (47.7)	184 (54.3) *	1079 (46.6)	69 (48.9)	580 (49.7)
ESI, n (%)	**		**			
1	4 (0.8)	48 (1.4)	2 (0.6)	17 (0.7)	2 (1.4)	31 (2.7)
2	123 (25.7)	705 (20.3)	78 (23.1)	342 (14.8)	45 (32.1)	363 (31.2)
3	257 (53.8)	1281 (36.9)	177 (52.4)	714 (30.9)	80 (57.1)	567 (48.7)
4	88 (18.4)	1317 (37.9)	76 (22.5)	1133 (49.0)	12 (8.6)	184 (15.8)
5	6 (1.3)	125 (3.6)	5 (1.5)	105 (4.5)	1 (0.7)	20 (1.7)

* *p* < 0.05, ** *p* < 0.001, p-value refers to the comparison between abdominal pain and no abdominal pain groups. ESI (Emergency Severity Index) category is the urgency level assigned at triage.

**Table 2 jcm-09-00899-t002:** Distribution of diagnoses.

All (*n* = 556)		<65 Years (*n* = 377)		≥65 Years (*n* = 179)	
Diagnosis	n (%)	Diagnosis	n (%)	Diagnosis	n (%)
Gastroenteritis	60 (10.8)	Gastroenteritis	54 (15.1)	Diverticulitis	13 (7.3)
NSAP *	58 (10.4)	NSAP *	53 (14.1)	Cholelithiasis	10 (5.6)
Cholelithiasis	25 (4.5)	*Urolithiasis*	22 (5.8)	*Urinary retention*	9 (5.0)
*Urolithiasis*	24 (4.3)	Appendicitis	19 (5.0)	Gastroenteritis	8 (4.5)
Diverticulitis	21 (3.8)	Cholelithiasis	15 (4.0)	Small bowel obstruction	8 (4.5)
Appendicitis	21 (3.8)	*Urinary tract infection*	12 (3.2)	*Urinary tract infection*	8 (4.5)
Small bowel obstruction	18 (3.2)	Disorders of ovary	10 (2.7)	*Influenza*	6 (3.4)
*Urinary tract infection*	17 (3.1)	Small bowel obstruction	10 (2.7)	Constipation	5 (2.8)
*Upper respiratory infection*	13 (2.3)	*Upper respiratory infection*	9 (2.4)	Gastrointestinal hemorrhage	5 (2.8)
*Urinary retention*	12 (2.2)	Diverticulitis	8 (2.1)	NSAP *	5 (2.8)
Disorders of ovary	10 (1.8)	*Pyelonephritis*	8 (2.1)	*Pneumonia*	5 (2.8)
GERD **	9 (1.6)	Pancreatitis	7 (1.9)	*Upper respiratory infection*	4 (2.2)
Gastrointestinal hemorrhage	9 (1.6)	GERD **	6 (1.6)	*Sepsis*	4 (2.2)
*Influenza*	9 (1.6)	*Somatoform disorders*	6 (1.6)	*COPD ****	3 (1.7)
Pancreatitis	8 (1.4)	*Lower back pain*	5 (1.3)	*Fracture rib(s), sternum*	3 (1.7)
*Pyelonephritis*	8 (1.4)	Gastrointestinal hemorrhage	4 (1.1)	GERD **	3 (1.7)
Constipation	7 (1.3)	Pregnancy	4 (1.1)	*Aortic aneurysm*	2 (1.1)
*Lower back pain*	7 (1.3)	Constipation	3 (0.8)	Appendicitis	2 (1.1)
*Pneumonia*	7 (1.3)	Endometriosis	3 (0.8)	*Epilepsy*	2 (1.1)
*Sepsis*	7 (1.3)	*Functional Dyspepsia*	3 (0.8)	Fibrosis and cirrhosis of liver	2 (1.1)
	350 (63)		262 (70)		107 (60)

* Non-specific abdominal pain. ** Gastro-esophageal reflux disease. *** Chronic obstructive pulmonary disease. *Italic type: Extra-abdominal diagnoses.* Diagnoses are ordered by number and alphabetically. Recording of up to two diagnoses per patient is possible.

**Table 3 jcm-09-00899-t003:** Missed final diagnoses.

Final Diagnosis (D)	Total, n (%)	D Correct at Index Visit, n	D Missed at Index Visit, n
Cholelithiasis	8 (13)	6	2
Gastroenteritis	7 (11)	2	5
Urinary retention	4 (6)	4	0
NSAP	3 (6)	2	1
Appendicitis	2 (3)	1	1
Constipation	2 (3)	1	1
Disorders of ovary	2 (3)	1	1
Diverticulitis	2 (3)	2	0
Endometriosis	2 (3)	2	0
Food intolerance	2 (3)	0	2
Malignant diseases	2 (3)	0	2
Pyelonephritis	2 (3)	2	0
Urolithiasis	2 (3)	1	1
Others	23 (37)	13	10
	63 (100)		

Missed diagnosis: final diagnosis at representation different from diagnosis at index visit.

**Table 4 jcm-09-00899-t004:** Outcomes.

	All (*n* = 3960)		<65 Years (*n* = 2652)		≥65 Years (*n* = 1308)	
	Abdominal Pain	No Abdominal Pain	Abdominal Pain	No Abdominal Pain	Abdominal Pain	No Abdominal Pain
Hospitalization, n (%)	170 (35.4) *	1067 (30.7)	87 (25.7) **	384 (16.6)	83 (58.9)	683 (58.5)
LOS (days), median (Q1, Q3)	5 (2.5, 9)	5 (2, 10)	4 (2, 7)	4 (2, 9)	7 (3, 11)	5 (2, 10)
ICU, n (%)	19 (5)	200 (5.8)	9 (2.7)	78 (3.4)	10 (7.1)	122 (10.5)
Mortality (in-hospital), n (%)	6 (1.3)	49 (1.4)	1 (0.3)	6 (0.3)	5 (3.6)	43 (3.7)
Mortality (30-day), n (%)	10 (2.2)	69 (2.1)	1 (0.3)	8 (0.4)	9 (6.6)	61 (5.4)
Mortality (1-year), n (%)	26 (5.8)	189 (5.8)	5 (1.6)	19 (0.9)	21 (15.3)	170 (15)

* *p* < 0.05, ** *p* < 0.001, *p*-value refers to comparison between groups abdominal pain and no abdominal pain groups. LOS (length of stay) was defined as the number of days spent in hospital during the index hospitalization. ICU (intensive care unit) was defined as any admission to medical, surgical ICU, or stroke unit. In-hospital mortality was defined as the percentage of patients who died during index hospitalization.

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
