# Peer review of "Acute Abdominal Pain: Missed Diagnoses, Extra-Abdominal Conditions, and Outcomes"

_jcm, 2020, doi:10.3390/jcm9040899_

Round 1

Reviewer 1 Report

INTRODUCTION

The Introduction could benefit from more exploration of the existing literature on AP - at the moment the Introduction is really only one paragraph; the rest of the Introduction then moves into information that is more pertinent to the Methods. The inclusion of additional information in the Introduction would also provide a stronger rationale for your hypotheses, such as that prognosis is highly dependent on age.

Some of the hypotheses require greater explication, particularly hypotheses 1 and 2:

  1. missed diagnoses leading to representation are considerable
  2. extra-abdominal causes of abdominal pain are frequent

What do you mean by 'considerable' and 'frequent'? How did you decide whether these hypotheses were met?

METHODS

P. 2 - lines 54-55 - Why were the two different time points chosen? Can you provide a rationale?

P. 2, line 55 - 'Conduction' should be 'conduct'

RESULTS

Figure 1 requires more explanation in the text. Similarly, it would be useful for the statistically significant results presented in Table 1 to be explained in the text.

P. 6, lines 185-186 - you state that "Older patients had a higher risk of ICU transfer: 10 (7.1%) versus 122 (10.5%) in the AP and non-AP subset, respectively", however, according to Table 3, this result is not statistically significant. I think you need to make it clearer in your explanation of the results in Table 3 that many are not statistically significant.

Overall, it would be useful to explicate in more detail the implications and significance of this work, for clinical practice and for future research. For instance, you might want to explore in more detail your suggestion that "protocol-based work-up needs to cover a considerable number and range of extra-abdominal conditions".

Reviewer 2 Report

Journal of Clinical Medicine

Acute Abdominal Pain: Missed Diagnoses, Extra-abdominal Conditions, and Outcomes

Peer Review Assessment

Congratulations to the authors on selecting a unique area for study. As a general surgeon, I frequently see patients in the emergency department for abdominal pain. I found this study very interesting and the data presented are of interest to a broad readership. This is a well-designed study and offers new and important findings. This study reminded me of a lesson from a classic surgical textbook – Cope’s Early Diagnosis of the Acute Abdomen: acute abdominal pain is simply abdominal pain that is sudden and severe enough to cause a patient to seek emergency medical evaluation. Acute abdominal pain does NOT always equate to surgical disease. As the authors have very nicely demonstrated, abdominal pain is not always from an abdominal source and the differential must always remain comprehensive.

Abstract

--I am not certain I understand the meaning of “representation” as used in your abstract and manuscript. Can you further clarify what this means in clinical context? Is this a second presentation to the emergency department? If so, in what time frame?

--With regards to mortality, can you clarify what the third number in parenthesis (6.6%) means?  

Introduction

--Your second question answered by your study relates to prevalence of extra-abdominal causes of abdominal pain. If you are looking at the measure of the probability of occurrence in a population within a specified period of time, do you actually mean to understand the incidence (not prevalence) of extra-abdominal causes of abdominal pain? This appears to be a study looking at cases over a specific period of time as opposed to a cross-sectional study. Please clarify this point in your manuscript.

Methods

--Did an Institutional Review Board approve this study as human subject research?  If questionnaires/survey research was conducted, IRB approval should have been obtained. This should be explicitly stated in the body of the manuscript.

--Were the medical students who worked 24/7 to enroll patients volunteer staff or was their participation required as part of their studies or clinical experience?

Results

--Line 181 – please clarify discharged from emergency department – some readers may interpret this as admitted and discharged from the hospital after admission.

Conclusions

--Is 13.1% the rate of patients who presented again to the emergency department during the study period? This seems very high and highlights an important area for investigation. How can we decrease the number of patients presenting again to the emergency department—especially for a problem related to their prior visit?

--Is 5.6% a low rate of missed diagnosis? This seems high to me. Can you put this missed diagnosis rate in context?

--You mention a 50% likelihood of missing a malignancy…can you please clarify or comment further.
